# Semantic-tailored Variational-Contrastive Graph Learning for Cognitive Diagnosis

Chenao Xia
Hefei University of Technology
Hefei, China
chenaoxia168@gmail.com

Fei Liu*
Hefei University of Technology
Hefei, China
feiliu@mail.hfut.edu.cn

Zihan Wang
Hefei University of Technology
Hefei, China
zhwang.hfut@gmail.com

Zhuangzhuang He
Hefei University of Technology
Hefei, China
hyicheng223@gmail.com

Pengyang Shao
Hefei University of Technology
Hefei, China
shaopymark@gmail.com

Haowei Zhou
Hefei University of Technology
Hefei, China
eric.hw.zhou@gmail.com

Yonghui Yang
Hefei University of Technology
Hefei, China
yyh.hfut@gmail.com

## Abstract

Cognitive diagnosis (CD), the foundation of intelligent education, aims to assess students' cognitive levels in knowledge concepts. Graph-based CD enhances diagnostic performance by incorporating high-order relations among entities, such as students, exercises, and knowledge concepts. Recently, self-supervised learning has been applied to CD to address data sparsity. However, existing contrastive learning methods may distort the student-exercise graph and overlook important semantic heterogeneity between correct and incorrect response logs. To address these limitations, we propose the *Semantic-tailored Variational-Contrastive Graph Cognitive Diagnosis (SVGCD)* method. First, a semantic-aware GNN is used to generate entity representations for different semantic environments. Then, a semantic-specific variational graph reconstruction module infers representation distributions and reconstructs semantic subgraphs while preserving the original graph structure. Additionally, a semantic-specific contrastive strategy introduces high-quality self-supervised signals while retaining semantic characteristics, enhancing student modeling for CD. Extensive experiments on two real-world datasets validate the effectiveness of our *SVGCD*. The code is available at https://github.com/XChuckie/SVGCD.

## CCS Concepts

• **Information systems → Information systems applications**;
• **Applied computing → Education**.

*Corresponding authors.

*WWW Companion '25, Sydney, NSW, Australia.*
© 2025 Copyright held by the owner/author(s). Publication rights licensed to ACM.
ACM ISBN 979-8-4007-1331-6/25/04
https://doi.org/10.1145/3701716.3717749

## Keywords

Intelligent Education, Cognitive Diagnosis, Graph Contrastive Learning, Student Performance Prediction

**ACM Reference Format:**
Chenao Xia, Fei Liu, Zihan Wang, Zhuangzhuang He, Pengyang Shao, Haowei Zhou, and Yonghui Yang. 2025. Semantic-tailored Variational-Contrastive Graph Learning for Cognitive Diagnosis. In *Companion Proceedings of the ACM Web Conference 2025 (WWW Companion '25), April 28-May 2, 2025, Sydney, NSW, Australia.* ACM, New York, NY, USA, 7 pages. https://doi.org/10.1145/3701716.3717749

## 1 Introduction

Intelligent education aims to improve the quality and equity of education through artificial intelligence[25, 28]. As its foundation, cognitive diagnosis (CD) assesses students' proficiency level in knowledge concepts, enabling the realization of AI4Education. As shown in Figure 1(a), diagnosis results of CD models enable various applications such as computerized adaptive testing [32] and exercise recommendation [14, 22]. Over the past decades, significant progress has been made in advancing cognitive diagnosis. Traditional models [2, 4, 18] use manually designed interaction functions to estimate student abilities, while neural network-based models [23, 24] leverage multi-layer perceptrons for improved generalizability and interpretability. Building on this, graph neural networks (GNNs)[12] have emerged to improve the accuracy and effectiveness of cognitive diagnosis by capturing high-order relationships between entities[6, 7, 10, 17, 30].

However, these studies overlook an important and rarely explored issue, ***data sparsity*** (i.e., students have very few response logs). And we carry out a data sparsity experiment using representative models (IRT [4], NeuralCD [23] and RCD [6]) to illustrate its harmful effects on CD by dividing student response logs into sparse, medium and dense groups in the ASSIST dataset [5]. As shown in Figure 1(b), *we observe that all models perform poorly in sparse student groups*. We attribute that students with fewer response logs are not adequately trained, resulting in their sub-optimal performance.

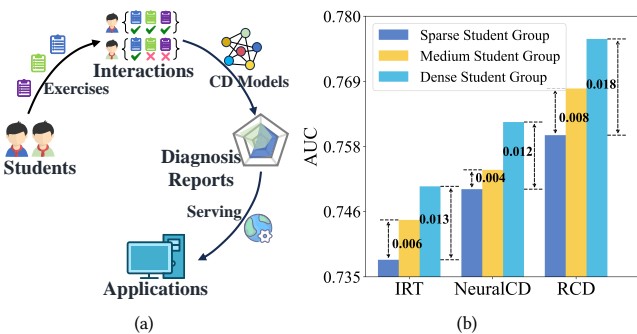

(a)

(b)

**Figure 1: (a) Pipeline of intelligent education system and (b) Performance comparisons of CD models over different students' groups (i.e., sparse, medium and dense groups).**

A few recent studies [26, 27, 31] have aimed to embrace the issue of data sparsity, and these methods are delving into the introduction of additional self-supervised signals on graph-based CD harnessing the potential of graph contrastive learning (GCL) [19, 29]. In addition to the supervised diagnosis task, GCL-based CD models first construct different contrastive views, and then maximize consistency between multiple view nodes, thus alleviating data sparsity while improving the performance of cognitive diagnosis.

Despite the notable performance of GCL-based CD models, we argue that there are still two limitations. **L1) Disrupt intrinsic nature of the student-exercise response graph**. Some models based on structural disturbances [26] randomly drop edges to build contrastive views for contrastive learning. Due to the fact that all nodes in the response graph are interrelated and do not satisfy the IID assumption, this strategy is easy to disrupt students' proficiency level on special concepts and intrinsic relationships among nodes, limiting the potential ability of contrastive learning. **L2) Lack in-depth consideration of semantic heterogeneity between correct and wrong student-exercise logs**. Current semantic-aware CD models [17, 20, 21] have proven remarkable performance, which is attributed to the consideration of semantic heterogeneity. And some GCL-based CD methods [27] attempt to model node representations of each semantic subgraph, and then directly adopt contrastive learning to pull the same nodes closer of both semantic subgraphs. Due to the significant differences of students' proficiency levels between the correct and wrong semantic environment, pulling the same nodes closer from different semantic subgraphs together can lead to conflicting information for the same student on the same knowledge concepts, causing unstable student ability modeling. Thus, it is crucial to carefully consider semantic heterogeneity in CD and design a reasonable contrastive strategy.

To address the above limitations, we propose a novel method, *Semantic-tailored Variational-Contrastive Graph Cognitive Diagnosis (SVGCD)*. First, we design a knowledge-integrated function to initialize student abilities and exercise difficulties by embedding knowledge information. Next, a semantic-aware GNN is employed to generate node representations from two semantic subgraphs (e.g., correct and wrong responses). For L1, we introduce a semantic-specific variational graph reconstruction module to infer node distributions for each semantic subgraph and reconstruct them with

the same topology but adjusted edge weights. For L2, we propose a semantic-specific contrastive learning strategy, which uses multiple sampling techniques to generate contrastive views based on semantic distributions, preserving semantic characteristics and mitigating data sparsity. Finally, for the supervised diagnosis task, we fuse node representations from both semantic environments as inputs to a cognitive diagnostic module for predicting student performance. The main contributions of this paper are as follows:

- We propose a novel self-supervised cognitive method *SVGCD*, which designs a semantic-specific variational graph reconstruction module to generate contrastive views without affecting the original graph structure.
- We design a semantic-specific contrastive strategy, using reconstructed subgraphs with the same semantics for contrastive learning to preserve their semantic characteristics.
- Extensive experiments and in-depth analysis on both datasets demonstrate the effectiveness of our method.

## 2 The Proposed Model

In this section, we introduce our proposed *Semantic-tailored Variational-Contrastive Graph Cognitive Diagnosis (SVGCD)* model. As illustrated by Figure 2, it aims to enhance graph-based cognitive diagnosis tasks with self-supervised learning.

**Preliminaries.** In the fundamental CD task, there are three sets of entities: students $\mathcal{S}(|\mathcal{S}| = N)$, exercises $\mathcal{V}(|\mathcal{V}| = M)$, and associated knowledge concepts $\mathcal{K}(|\mathcal{K}| = C)$. Exercise-knowledge mapping matrix $\mathbf{Q} = \{q_{vk}\}_{M \times C}$ can be regarded as a binary matrix labeled by domain experts, where $q_{vk} = 1$ if exercise $v$ relates to concept $k$, otherwise $q_{vk} = 0$. The students' response logs are a triplet set $R = \{(s, v, r_{sv}) | s \in \mathcal{S}, v \in \mathcal{V}, r_{sv} \in \{0, 1\}\}$, where $r_{sv} = 1$ means a correct answer while $r_{sv} = 0$ means an wrong answer. In this paper, we utilize response logs to build a bipartite graph $\mathcal{G} = \{\mathcal{S} \cup \mathcal{V}, \mathbf{A}\}$, where $\mathcal{S} \cup \mathcal{V}$ involves all students and exercises, and $\mathbf{A}$ denotes the adjacency matrix defined by training response logs. Due to both different semantics of response logs, we segment the original graph into two subgraphs: correct semantic subgraph $\mathcal{G}_{(+)} = \{\mathcal{S} \cup \mathcal{V}, \mathbf{A}_{(+)}\}$ and wrong semantic subgraph $\mathcal{G}_{(-)} = \{\mathcal{S} \cup \mathcal{V}, \mathbf{A}_{(-)}\}$. Based on this, our goal is to mine students' cognitive states on specific knowledge concepts by predicting the scores on related exercises.

### 2.1 Semantic-aware Graph Cognitive Diagnosis

Before predicting students' scores, we project entities into a $d$-dimensional embedding space. Specifically, let $\mathbf{U}^{(\mathcal{S})} \in \mathbb{R}^{N \times d}$, $\mathbf{B}^{(\mathcal{V})} \in \mathbb{R}^{M \times d}$ and $\mathbf{D}^{(\mathcal{K})} \in \mathbb{R}^{C \times d}$ represent the embeddings of students, exercises and knowledge concepts respectively. We generate embedding vectors $\boldsymbol{u}_s$, $\boldsymbol{b}_v$ and $\boldsymbol{d}_k$ of size $\mathbb{R}^d$ for student $s$, exercise $v$ and concept $k$.

*2.1.1 Knowledge-Integrated Initialization.* To capture interpretable student and exercise traits on knowledge concepts within the student-exercise bipartite graph, we design a Knowledge-Integrated Function (KIF), inspired by the success of implicitly modeling intrinsic knowledge relations [15, 20, 24], as follows:

$$\theta_{sk}^0 = \boldsymbol{u}_s \times (\boldsymbol{d}_k)^T, \quad \psi_{vk}^0 = \boldsymbol{b}_v \times (\boldsymbol{d}_k)^T, \tag{1}$$

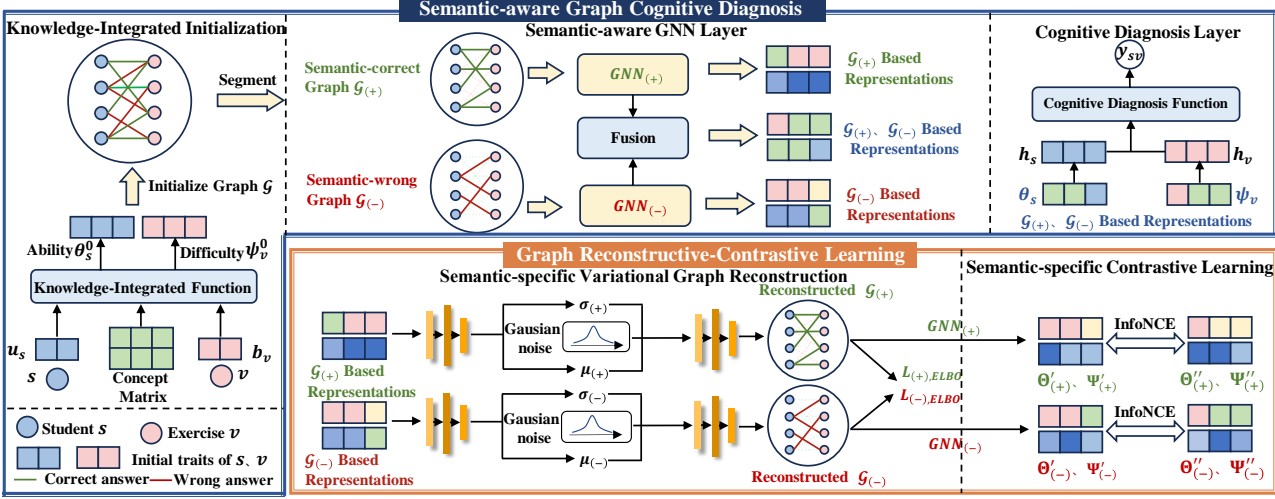

Figure 2: The overall structure of our proposed *SVGCD*.

where $\theta^0_{sk}$, $\psi^0_{vk}$ denote raw proficiency of $s$ on $k$ and raw difficulty of $v$ on $k$, respectively. We generate the ability vector $\boldsymbol{\theta}^0_s \in \mathbb{R}^{1 \times C}$ and the difficulty vector $\boldsymbol{\psi}^0_v \in \mathbb{R}^{1 \times C}$ for student $s$ and exercise $v$. And we obtain embedding matrices $\Theta^0 \in \mathbb{R}^{N \times C}$ and $\Psi^0 \in \mathbb{R}^{M \times C}$ of all students and exercises.

*2.1.2 Semantic-aware GNN Layer.* Given both semantic subgraphs $\mathcal{G}_{(+)}$ and $\mathcal{G}_{(-)}$, employing GNNs can capture the high-order relations [6] within entities. At the core is, for initialized node embeddings $\mathbf{E}^0 = [\Theta^0, \Psi^0]$, we use lightweight graph convolutional networks [9] to update the representation of ego nodes of both subgraphs $\mathcal{G}_{(+)}$ and $\mathcal{G}_{(-)}$:

$$\mathbf{E}^l_{(*)} = \mathbf{D}^{-\frac{1}{2}}_{(*)} \mathbf{A}_{(*)} \mathbf{D}^{-\frac{1}{2}}_{(*)} \mathbf{E}^{l-1}_{(*)}, \ * \in \{+, -\}, \quad (2)$$

where $\mathbf{E}^l_{(*)}$ and $\mathbf{E}^{l-1}_{(*)}$ denote the node representations in $l^{th}$ and $(l-1)^{th}$ graph convolution layer. $\mathbf{D}_{(*)}$ is diagonal degree matrices of semantic subgraphs. Additionally, we obtain the semantic-aware fused representation with an intra-layer additive strategy to server inputs of cognitive diagnosis layer: $\mathbf{E}^l_{(f)} = \mathbf{E}^l_{(+)} + \mathbf{E}^l_{(-)}$. After obtaining all representations of $L$ layers, the final representations are obtained with a readout function:

$$\mathbf{E}_{(*)} = f_{readout}(\mathbf{E}^0_{(*)}, \ \mathbf{E}^1_{(*)}, \ \ldots, \ \mathbf{E}^L_{(*)}), \ * \in \{+, -, f\}, \quad (3)$$

where $f_{readout}$ can be mean, sum, concatenation, or last-layer output. For brevity, we summarize the procedure of semantic-aware GNN on semantic subgraphs $\mathcal{G}_{(*)}$ to obtain aggregated node representations, formulated as:

$$\Theta_{(*)}, \Psi_{(*)} = \text{GNN}_{(*)}(\mathcal{G}_{(*)} | \mathbf{E}^0_{(*)}), \ * \in \{+, -, f\}, \quad (4)$$

where $\Theta_{(*)} \in \mathbb{R}^{N \times C}$ and $\Psi_{(*)} \in \mathbb{R}^{M \times C}$ denote different forms of final student and exercise representations in both semantic environments: correct, wrong, and fused semantic embedding.

*2.1.3 Cognitive Diagnosis Layer.* After obtaining semantic-aware fused embedding vectors $\theta_s \in \Theta_f$ and $\psi_v \in \Psi_f$ of student $s$ and exercise $v$, we adopt a single-layer perception with parameter constraints, which meet monotonicity conditions [4, 18, 23], to learn the diagnosis state of $s$ and $v$, formulated as:

$$\boldsymbol{h}_s = \sigma(W_s \times \boldsymbol{\theta}_s + \boldsymbol{b}_s), \ \boldsymbol{h}_v = \sigma(W_v \times \boldsymbol{\psi}_v + \boldsymbol{b}_v), \quad (5)$$

where $\sigma(\cdot)$ represents the sigmoid activation function. $W_s$ and $W_v$ are the weights of neural networks, which ensure the interpretability of diagnosis results. To verify the correctness of the diagnosis, we use the NeuralCD paradigm [23] to map the diagnosis results into predicted response logs:

$$y_{sv} = \sigma(\text{MLPs}(\mathbf{Q}_v \odot (\boldsymbol{h}_s - \boldsymbol{h}_v) \times \boldsymbol{h}^{disc}_v)), \quad (6)$$

where $\odot$ denotes the element-wise product. MLPs denotes multi-layer perceptions with positive constraints and $\boldsymbol{h}^{disc}_v$ denotes exercise discriminations. Finally, we utilize the cross entropy between predicted score $y_{sv}$ and true score $r_{sv}$ as the loss function of cognitive diagnosis, defined as:

$$\mathcal{L}_{cd} = - \sum_{(s,v,r_{sv}) \in R} (r_{sv} \log y_{sv} + (1 - r_{sv}) \log(1 - y_{sv})), \quad (7)$$

## 2.2 Graph Reconstructive-Contrastive Learning

Existing GCL-based CD models [26, 27] can destroy important intrinsic properties of student-practice graphs and fail to fully consider semantic heterogeneity, limiting the potential of contrastive learning. Therefore, we propose semantics-specific variational graph reconstruction [13] to customise nodes' distribution of each semantic environment, and then generate contrastive views without damaging the original topology structure. Further, *SVGCD* employs semantics-specific contrastive strategy, divide-and-conquer thinking, which effectively preserves the characteristics of different semantics while alleviating data sparsity.

*2.2.1 Semantic-specific Variational Graph Reconstruction.* After obtaining the nodes' representations of the semantic environments

$E_{(+)}$ and $E_{(-)}$, we build a variational graph reconstruction module that uses probability distributions $\mathcal{Z}$ to reconstruct the origin semantic subgraphs without destroying the original topology: $\hat{\mathcal{G}}_{(*)} \sim p_\theta(\mathcal{G}_{(*)}|\mathcal{Z}_{(*)})$, where $\mathcal{G}_{(*)} = \{\mathcal{G}_{(+)}, \mathcal{G}_{(-)}\}$ denotes two semantic subgraphs. Further, we define the reconstructive distribution as $p(\mathcal{G}_{(*)}) = \int p_\theta(\mathcal{G}_{(*)}|\mathcal{Z}_{(*)})p(\mathcal{Z}_{(*)})d\mathcal{Z}_{(*)}$. However, integration on $\mathcal{Z}$ is intractable due to unknown probability distributions $\mathcal{Z}$. Thus, we utilize the reparameterization trick [11] and apply variational inference to optimize the Evidence Lower Bound (ELBO):

$$\log p(\mathcal{G}_{(*)}) = \log \int p_\theta(\mathcal{G}_{(*)}|\mathcal{Z}_{(*)})p(\mathcal{Z}_{(*)})d\mathcal{Z}_{(*)}$$
$$\geq \mathbb{E}_{q_\phi}\left[\log p_\theta(\mathcal{G}_{(*)}|\mathcal{Z}_{(*)})\right] - KL\left[q_\phi(\mathcal{Z}_{(*)}|\mathcal{G}_{(*)})||p(\mathcal{Z}_{(*)})\right], \quad (8)$$

where $q_\phi(\mathcal{Z}_{(*)}|\mathcal{G}_{(*)})$ and $p_\theta(\mathcal{G}_{(*)}|\mathcal{Z}_{(*)})$ denote the variational inference encoder and graph reconstruction decoder, which are parameterized by neural networks. $KL$ presents KL-divergence between the approximate posterior $q_\phi(\mathcal{Z}_{(*)}|\mathcal{G}_{(*)})$ and prior $p(\mathcal{Z}_{(*)})$. Then we describe the two neural networks mentioned above.

*Variational Inference Encoder.* To understand the intrinsic distribution of each node and customize contrastive views of each semantic environment, we use a variational encoder to estimate the distributions of the nodes of each semantic subgraph: $q_\phi(\mathcal{Z}_{(*)}|\mathcal{G}_{(*)}) = \prod_{i=0}^{N+M-1} q_\phi(\mathcal{Z}_{(*),i}|G_{(*)})$. Futher, we encode each node $i$ of the semantic subgraphs $\mathcal{G}_{(*)}$ into the multi-variate Gaussian distributions: $q_\phi(\mathbf{z}_{(*),i}|\mathcal{G}_{(*)}) = \mathcal{N}(\mathbf{z}_{(*),i}|\boldsymbol{\mu}_{(*),\phi}(i), diag(\sigma^2_{(*),\phi}(i)))$, where $\boldsymbol{\mu}_{(*),\phi}(i)$ and $\sigma^2_{(*),\phi}(i)$ denote the mean and variance of the node $i$ distribution. Considering the implicit correlation between the nodes on each semantic subgraph, we use outputs of semantic subgraphs $\mathcal{G}_{(*)}$: $E_{(*)} = [\mathbf{E}^0_{(*)}, \mathbf{E}^1_{(*)}, ..., \mathbf{E}^L_{(*)}]$, to evaluate the means and variances of each node, formulated as:

$$\boldsymbol{\mu}_{(*)} = \frac{1}{L}\sum_{l=1}^{L}\mathbf{E}^l_{(*)}, \quad \boldsymbol{\sigma}_{(*)} = \text{MLPs}(\boldsymbol{\mu}_{(*)}), \quad (9)$$

where MLPs represents two linear transformation functions. After obtaining the mean and variance of the approximate posterior distribution, we sample a latent representation $\mathbf{z}_{(*),i}$, which followed the isotropic Gaussian distribution $p(\mathcal{Z}_{(*)}) \sim \mathcal{N}(\boldsymbol{\mu}_{(*)}, \sigma^2_{(*)}I)$, for each node $i$ of $\mathcal{G}_{(*)}$. However, direct sampling makes it difficult to calculate gradients for the entire model, as the sampling process is non-differentiable. Thus, we adopt the reparameterization trick [11] as an alternative to sampling from the distribution $\mathcal{N}(\boldsymbol{\mu}_{(*),i}, \sigma^2_{(*),i}I)$, formulated as follows:

$$\mathbf{z}_{(*),i} = \boldsymbol{\mu}_{(*),i} * \boldsymbol{\sigma}_{(*),i} \odot \boldsymbol{\varepsilon}, \quad (10)$$

where $\boldsymbol{\varepsilon} \sim \mathcal{N}(0, \mathbf{I})$ is a standard Gaussian distribution.

*Graph Reconstruction Decoder.* After estimating latent probability distribution of nodes of various semantic subgraphs $\mathcal{G}_{(+)}$ and $\mathcal{G}_{(-)}$: $\mathcal{Z}_{(*)} = \{\mathcal{Z}_{(+)}, \mathcal{Z}_{(-)}\}$, we obtain reconstructive distributions of origin semantic subgraphs: $p_\theta(\mathcal{G}_{(+)}|\mathcal{Z}_{(+)})$ and $p_\theta(\mathcal{G}_{(-)}|\mathcal{Z}_{(-)})$. Thus reconstructed subgraphs $\hat{\mathcal{G}}_{(*)}$ can be formulated as follows:

$$p_\theta(\mathcal{G}_{(*)}|\mathcal{Z}_{(*)}) = \prod_{i=0}^{N*M-1}\prod_{j=0}^{N*M-1} p_\theta(\mathbf{A}_{(*),ij}|\mathbf{z}_{(*),i}, \mathbf{z}_{(*),j}), \quad (11)$$

Considering the specificity of CD, that is, the difference between the student's ability and the difficulty of the exercise reflects the student's mastery of the exercise, we use the difference between reconstructed student $i$ and exercise $j$ to present the propensity score that student $i$ interacted with exercise $j$, as follows:

$$p_\theta(\mathbf{A}_{(*),ij}|\mathbf{z}_{(*),i}, \mathbf{z}_{(*),j}) = \sigma(f_\theta(\mathbf{z}_{(*),i} - \mathbf{z}_{(*),j})), \quad (12)$$

herein, $f_\theta(\cdot)$ is a simple but effective multi-layer perceptions parameterized by $\theta$.

*2.2.2 Semantic-specific Contrastive Learning.* After getting contrastive views of $\mathcal{G}_{(+)}$ by multiple samplings of the estimated distributions: $\hat{\mathcal{G}}'_{(+)}$ and $\hat{\mathcal{G}}''_{(+)}$, according to Eq. (10) and Eq. (11). We utilize $\text{GNN}_{(+)}(\cdot)$ like Eq. (4), to obtain students and exercises representations of two correct semantic reconstruction views:

$$\begin{aligned}\boldsymbol{\Theta}'_{(+)}, \boldsymbol{\Psi}'_{(+)} &= \text{GNN}_{(+)}(\hat{\mathcal{G}}'_{(+)}|\Omega),\\ \boldsymbol{\Theta}''_{(+)}, \boldsymbol{\Psi}''_{(+)} &= \text{GNN}_{(+)}(\hat{\mathcal{G}}''_{(+)}|\Omega),\end{aligned} \quad (13)$$

Following GCL-based CD paradigms [26], we adopt the InfoNCE [16] loss as an auxiliary objective to maximize the mutual information lower bound. This encourages consistency between representations of the same nodes across views while distinguishing representations of different nodes.

$$\begin{aligned}\mathcal{L}^{inv}_{(+),S} &= \sum_{a \in \mathcal{B}_{(+),s}} -\log \frac{\exp(\theta'^{T}_{(+),a}\theta''_{(+),a}/\tau)}{\sum_{b \in \mathcal{B}_{(+),s}}\exp(\theta'^{T}_{(+),a}\theta''_{(+),b}/\tau)},\\ \mathcal{L}^{inv}_{(+),E} &= \sum_{i \in \mathcal{B}_{(+),e}} -\log \frac{\exp(\psi'^{T}_{(+),i}\psi''_{(+),i}/\tau)}{\sum_{j \in \mathcal{B}_{(+),e}}\exp(\psi'^{T}_{(+),i}\psi''_{(+),j}/\tau)},\end{aligned} \quad (14)$$

where hyper-parameter $\tau$ is the temperature coefficient, $\mathcal{B}_{(+),s}$ and $\mathcal{B}_{(+),e}$ denote students and exercises from correct responses in batch training data. By combining two losses, we obtain the objective of contrastive learning of $\mathcal{G}_{(+)}$, denoted as $\mathcal{L}^{inv}_{(+)} = \mathcal{L}^{inv}_{(+),S} + \mathcal{L}^{inv}_{(+),E}$. Analogously, we compute the contrastive loss within the wrong response side as $\mathcal{L}_{(-)} = \mathcal{L}^{inv}_{(-),S} + \mathcal{L}^{inv}_{(-),E}$. Consequently, the final objective of contrastive learning task can represent the cumulative loss mentioned above: $\mathcal{L}_{cl} = \mathcal{L}_+ + \mathcal{L}_-$.

## 2.3 Model Optimization for *SVGCD*

For semantic-specific variational graph reconstruction part, we use ELBO loss mentioned in Eq. (8) to optimize the parameters of variational inference encoder and graph reconstruction decoder for various semantic reconstructed subgraphs $\hat{\mathcal{G}}_{(+)}$ and $\hat{\mathcal{G}}_{(-)}$:

$$\mathcal{L}_{(*),ELBO} = -\mathbb{E}_{q_\phi}\left[\log p_\theta(\mathcal{G}_{(*)}|\mathcal{Z}_{(*)})\right] + KL\left[q_\phi(\mathcal{Z}_{(*)}|\mathcal{G}_{(*)})||p(\mathcal{Z}_{(*)})\right], \quad (15)$$

Then, we get the final variational graph reconstruction loss $\mathcal{L}_{ELBO} = \mathcal{L}_{(+),ELBO} + \mathcal{L}_{(-),ELBO}$. Overall, the training of the proposed *SVGCD* adopts multi-task learning, consisting of three subparts as follows:

$$\min \mathcal{L} = \mathcal{L}_{cd} + \lambda\mathcal{L}_{cl} + \beta\mathcal{L}_{ELBO}. \quad (16)$$

where $\lambda$ and $\beta$ are hyperparameters that control the strengths of auxiliary tasks. During inference, we use Eq. (4) to compute the representations $\boldsymbol{\Theta}$ and $\boldsymbol{\Psi}$ without variational graph reconstruction.

**Table 1: Performance comparison at different train/test split ratios. The best performance is highlighted in bold, and the runner-ups are with underlines. ↑ (↓) denotes the higher (lower) score the better (worse) performance. By t-test, the bold result is statistically significantly better than others for each metric, with a significant level of $\alpha = 5\%$.**

| Metrics | | AUC ↑ | | | | RMSE ↓ | | | | ACC ↑ | | | |
|---|---|---|---|---|---|---|---|---|---|---|---|---|---|
| Datasets | Methods | 5:5 | 6:4 | 7:3 | 8:2 | 5:5 | 6:4 | 7:3 | 8:2 | 5:5 | 6:4 | 7:3 | 8:2 |
| ASSIST | IRT | 0.7225 | 0.7333 | 0.7415 | 0.7480 | 0.4472 | 0.4434 | 0.4412 | 0.4390 | 0.7062 | 0.7100 | 0.7153 | 0.7175 |
| | NeuralCD | 0.7340 | 0.7413 | 0.7508 | 0.7558 | 0.4440 | 0.4408 | 0.4386 | 0.4372 | 0.7131 | 0.7155 | 0.7151 | 0.7186 |
| | KaNCD | 0.7412 | 0.7499 | 0.7584 | 0.7640 | 0.4468 | 0.4399 | 0.4340 | 0.4304 | 0.7126 | 0.7214 | 0.7269 | 0.7315 |
| | RCD | 0.7563 | 0.7612 | 0.7644 | 0.7685 | 0.4287 | 0.4257 | 0.4255 | 0.4231 | 0.7256 | 0.7293 | 0.7305 | 0.7330 |
| | SCD | 0.7543 | 0.7602 | 0.7668 | 0.7701 | 0.4304 | 0.4276 | 0.4245 | 0.4233 | 0.7222 | 0.7262 | 0.7330 | 0.7321 |
| | ASG-CD | 0.7545 | 0.7631 | 0.7697 | 0.7736 | 0.4331 | 0.4268 | 0.4244 | 0.4219 | 0.7259 | 0.7313 | 0.7360 | 0.7371 |
| | **SVGCD (Ours)** | **0.7640** | **0.7713** | **0.7775** | **0.7832** | **0.4251** | **0.4213** | **0.4188** | **0.4158** | **0.7325** | **0.7362** | **0.7413** | **0.7434** |
| Junyi | IRT | 0.7867 | 0.7901 | 0.7927 | 0.7858 | 0.4171 | 0.4131 | 0.4103 | 0.4075 | 0.7281 | 0.7390 | 0.7454 | 0.7508 |
| | NeuralCD | 0.7803 | 0.7804 | 0.7806 | 0.7811 | 0.4129 | 0.4125 | 0.4122 | 0.4126 | 0.7473 | 0.7479 | 0.7476 | 0.7473 |
| | KaNCD | 0.7850 | 0.7859 | 0.7869 | 0.7885 | 0.4092 | 0.4091 | 0.4089 | 0.4083 | 0.7504 | 0.7509 | 0.7516 | 0.7556 |
| | RCD | 0.8035 | 0.8056 | 0.8066 | 0.8087 | 0.3997 | 0.3998 | 0.3984 | 0.3972 | 0.7666 | 0.7675 | 0.7673 | 0.7689 |
| | SCD | 0.8026 | 0.8073 | 0.8099 | 0.8134 | 0.4003 | 0.3981 | 0.3968 | 0.3949 | 0.7646 | 0.7676 | 0.7687 | 0.7726 |
| | ASG-CD | 0.8079 | 0.8105 | 0.8128 | 0.8156 | 0.3980 | 0.3964 | 0.3957 | 0.3938 | 0.7718 | 0.7725 | 0.7732 | 0.7754 |
| | **SVGCD (Ours)** | **0.8162** | **0.8189** | **0.8205** | **0.8233** | **0.3932** | **0.3924** | **0.3908** | **0.3892** | **0.7755** | **0.7759** | **0.7780** | **0.7804** |

**Table 2: Statistics of two real-world datasets for experiments.**

| Dataset | ASSIST | Junyi |
|---|---|---|
| # Students | 2,493 | 10,000 |
| # Exercises | 17,676 | 734 |
| # Knowledge concepts | 123 | 734 |
| # Response logs | 267,423 | 408,057 |
| # Avg logs per student | 107.240 | 40.806 |
| # Sparsity in student-exercise logs | 99.394% | 94.441% |

**Table 3: Ablation study of *SVGCD* on two datasets.**

| Datasets | | ASSIST | | | Junyi | |
|---|---|---|---|---|---|---|
| Metrics | AUC | RMSE | ACC | AUC | RMSE | ACC |
| SVGCD | **0.7832** | **0.4158** | **0.7434** | **0.8233** | **0.3892** | **0.7804** |
| w/o VG+w/ R | 0.7782 | 0.4199 | 0.7402 | 0.8188 | 0.3920 | 0.7768 |
| w/o DCL+w/ P | 0.7754 | 0.4201 | 0.7395 | 0.8182 | 0.3956 | 0.7763 |
| w/o SCL | 0.7702 | 0.4262 | 0.7349 | 0.8037 | 0.4033 | 0.7664 |

## 3 Experiments

### 3.1 Experimental Settings

To evaluate the cognitive performance of *SVGCD*, we conduct experiments on two public datasets: the ASSISTments 2009-2010 skill builder dataset [1] [5] and Junyi Academy Math Practicing Log [2] [1], containing student-exercise response logs and exercise-knowledge relational matrices. Following prior work [6, 26], we retain only the first attempt per question and filter out students with fewer than 15 responses. For the Junyi dataset, we randomly select 10,000 response logs. Finally, we evaluate our method using various train/test splits. The dataset statistics are summarized in Table 2.

We evaluate the effectiveness of *SVGCD* by comparing it with six well-known CD models, **IRT** [4], **NeuralCD** [23], **KaNCD** [24], **RCD** [6], **SCD** [26], **ASG-CD** [20]. Performance is measured using three metrics: ACC, RMSE, and AUC. All models, including *SVGCD*, are implemented in PyTorch. Model parameters are initialized with Xavier initialization [8], embedding size is set to 128, and training

---

[1]https://sites.google.com/site/assistmentsdata/feng2009
[2]https://pslcdatashop.web.cmu.edu/DatasetInfo?datasetId=1198

uses Adam [3] with a learning rate of 0.0001 and batch size of 1024. For *SVGCD*, we tune key hyperparameters: contrastive temperature $\tau$ (0.3–1.0), GNN layers (1–5), contrastive loss $\beta$, and variational loss $\lambda$ (0.1–1.0). Baseline parameters are selected as per their respective papers. All experiments are repeated five times, and average results are reported, using a GeForce RTX 3090 GPU.

### 3.2 Experimental Results

**Overall Comparison.** To demonstrate the superiority of *SVGCD*, we compare it with well-known methods on two popular datasets using three standard metrics. Table 1 summarizes the results of all models, and the variance of each metric is less than 0.05. First: *SVGCD* consistently outperforms all baselines. Across both datasets, it achieves significantly better results ($\alpha = 5\%$) for all metrics, regardless of train/test split ratios, confirming its effectiveness and the importance of well-designed contrastive views in graph cognitive diagnosis. Second: *SVGCD* shows notable improvement over GCL-based CD models. Compared to SCD, which employs structural augmentation for contrastive views, *SVGCD* demonstrates substantial gains, thanks to its semantic-specific variational graph reconstruction and novel contrastive strategy. These modules infer

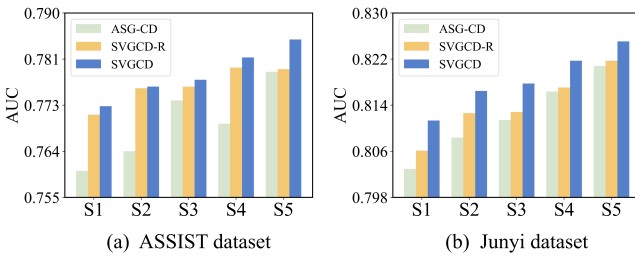

(a) ASSIST dataset      (b) Junyi dataset

**Figure 3: Performance** $w.r.t.$ **different student groups.**

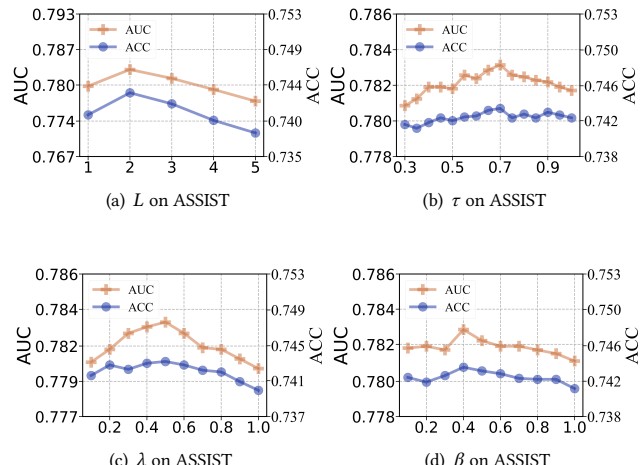

(a) $L$ on ASSIST      (b) $\tau$ on ASSIST

(c) $\lambda$ on ASSIST      (d) $\beta$ on ASSIST

**Figure 4: Performance of different hyperparameters.**

node distributions and generate rational contrastive views, preserving both structural integrity and the unique characteristics of distinct semantic environments.

**Ablation Studies.** The experiments above demonstrate the overall performance of *SVGCD*. To further analyze its components, we evaluate *SVGCD* on both datasets using an 8:2 train/test split (Table 3). Our model outperforms the variant "w/o VG + w/ R", where the variational module is replaced with random augmentation, confirming that the variational module generates superior contrastive views without structural distortion. To assess the importance of semantic heterogeneity, we replace the proposed divide-and-conquer contrastive learning (DCL) strategy with a simpler approach that pulls the same nodes of both semantic subgraphs (variant "w/o DCL + w/ P") [27]. This variant performs significantly worse, highlighting the impact of proficiency differences between correct and wrong semantic environments. Finally, by removing semantic-specific contrastive learning (variant "w/o SCL"), the performance sudden drop can be attributed to the fact that contrastive learning can learn the invariant characteristic by maximizing the mutual information of different contrastive views. Thus, all components of SVGCD contribute to the final superior performance.

**Model Robustness to Data Sparsity.** We investigate the effectiveness of our method in mitigating data sparsity by comparing it with optimal baseline ASG-CD and SVGCD-R (used structural augmentation to generate contrastive views). Specifically, we split all students into five groups according to the ascending number of response logs in the training set in both datasets. Figure 3 illustrates the performance of various groups, we find that GCL-based CD methods significantly improve compared to ASG-CD in all student groups, especially in sparse student groups. This indicates that the introduction of auxiliary self-supervised signals facilitates the data sparsity issue. Further, our SVGCD consistently outperforms two methods in each group of students, better validating the effectiveness of the proposed contrastive strategy.

**Hyper-Parameter Sensitivities.** As shown in Figure 4, we experimentally analyze the sensitivity of *SVGCD* to four main hyperparameters on ASSIST dataset. First: GNN Layer $L$. We explore the parameter $L$ within the range of {1, 2, 3, 4, 5} as shown in Figure 4(a), the diagnosis performances initially improve with an increase of GNN layers, but slightly decrease when the number of layers exceeds two. It indicates that shallow layers fail to capture high-order relations among nodes, while overly deep layers also lead to the over-smoothing issue, thereby diminishing performance. Second: Temperature coefficient $\tau$. In Figure 4(b), we observe that an excessively high $\tau$ leads to poor performance, as the model fails to

effectively mine hard negative samples. Conversely, a too-low $\tau$ also hampers performance by making the model focus on false negative samples. Three: Contrastive learning weight $\lambda$ and variational reconstructive weight weight $\beta$. As illustrated in Figure 4(c) and Figure 4(d), we carefully tune the weight $\lambda$ and $\beta$, and then observe that our model shows a trend of first upward and then falling. Note that our method achieves optimal performance with $\lambda = 0.5$ and $\beta = 0.4$ on the ASSIST dataset. Thus, appropriate parameters can effectively mitigate data sparsity issues; in turn, improper parameter settings would lead to suboptimal diagnosis performance.

## 4 Conclusion

In this work, we proposed a novel semantic-tailored graph contrastive cognitive diagnosis method *SVGCD*. Specifically, we reconstruct specific semantic subgraphs without distorting the original graph structure by inferring the distribution of nodes with the same semantic information based on variational graph reconstruction techniques. Then we propose a divide-and-conquer strategy tailored for contrastive learning in the field of cognitive diagnosis, which generates multiple contrastive views with multiple samplings for each semantic environment based on estimated semantic distributions, thus preserving their semantic characteristics. Extensive experimental results on both datasets demonstrated the effectiveness of our proposed *SVGCD*.

## Acknowledgements

This work was supported in part by grants from the National Science and Technology Major Project (under Grant 2021ZD0111802), the National Natural Science Foundation of China (under Grant 62406096), the China Postdoctoral Science Foundation (under Grant 2024M760722), the Fundamental Research Funds for the Central Universities (under Grant JZ2024HGQB0093), and the Anhui Postdoctoral Scientific Research Program Foundation (under Grant No. 2024C934).

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
