# OpenReview forum: "Semantic-tailored Variational-Contrastive Graph Learning for Cognitive Diagnosis"
_ACM.org/TheWebConf/2025/Workshop/TIME — TIME 2025 Oral_

### Official Review · Reviewer_cLtg · 2025-01-08
**Semantic-tailored Variational-Contrastive Graph Learning for Cognitive Diagnosis**

**Rating:** 5
**Confidence:** 3

**Review:**

1.The architecture is notclear for instance how the model differentiates the good and bad semantic samples.
2.What is the complexity of the solution
3.How is this architectural solution effective and how effcicint as compared to other solution in the domain

---

### Official Review · Reviewer_qQFt · 2025-01-13
**Reviews of "Semantic-tailored Variational-Contrastive Graph Learning for Cognitive Diagnosis"**

**Rating:** 7
**Confidence:** 4

**Review:**

The paper proposes a Semantic Customized Variational Contrastive Graph Learning method (SVGCD) for cognitive diagnosis, aiming to address the issues of data sparsity and the limitations of existing contrastive learning methods in graph-based cognitive diagnosis.

Advantages:

- SVGCD is significant for advancing intelligent education.
- The experimental setup is reasonable, comparing various well-known CD models.
- SVGCD effectively addresses the issues of distorted student-exercise graph structures and insufficient consideration of semantic heterogeneity.

Disadvantages:

- The SVGCD model involves multiple hyperparameters, such as lambda and beta, whose impact on performance remains unknown.

---

### Official Review · Reviewer_SZRk · 2025-01-15
**Review for #6**

**Rating:** 6
**Confidence:** 1

**Review:**

I am not an expert in this field, but judging from the writing of the paper, it may have good quality.

The baseline results of the proposed method should also be compared in Table 3.

---

### Official Review · Reviewer_CMMm · 2025-01-16
**This review evaluates the Semantic-tailored Variational-Contrastive Graph Cognitive Diagnosis (SVGCD) method, highlighting its quality, clarity, originality, and significance. The paper presents a novel approach by integrating semantic-specific variational graph reconstruction with contrastive learning, demonstrating superior performance over existing models. While the methodology is innovative and well-supported by experiments, its complexity and potential generalizability to other domains warrant further discussion. Overall, SVGCD offers a promising advancement in cognitive diagnosis for intelligent education systems.**

**Rating:** 7
**Confidence:** 3

**Review:**

Quality:

The paper does a good job of laying out a new approach to cognitive diagnosis with a method called Semantic-tailored Variational-Contrastive Graph Cognitive Diagnosis (SVGCD). The authors explain their methodology clearly, focusing on how they use semantic-specific variational graph reconstruction and contrastive learning strategies. They also back up their claims with solid experiments, comparing SVGCD to six well-known cognitive diagnosis models using various metrics.

Clarity:

Overall, the paper is pretty clear. The introduction effectively highlights why cognitive diagnosis is important in intelligent education. The methodology section is detailed, explaining how semantic-aware graph cognitive diagnosis and semantic-specific contrastive learning work. However, some technical terms and processes might be a bit tough to grasp for readers who aren't familiar with graph-based learning methods.

Originality:
This work stands out because it integrates semantic-specific variational graph reconstruction with contrastive learning for cognitive diagnosis in a new way. The divide-and-conquer strategy for creating contrastive views based on semantic distributions is a fresh contribution to the field.

Significance:

The paper is significant because it could improve cognitive diagnosis models by keeping semantic characteristics intact and dealing with data sparsity. The results from the experiments show that SVGCD is effective, suggesting it could be a valuable tool in intelligent education systems.

Pros:
Innovative Approach: Combining variational graph reconstruction and contrastive learning tailored to semantic environments is a novel idea.
Comprehensive Evaluation: The model is tested against several established cognitive diagnosis models and shows better performance.
Focus on Semantic Characteristics: The method does a good job of preserving semantic characteristics, which is crucial for accurate cognitive diagnosis.

Cons:
Complexity: The methodology might be a bit complex for those who aren't familiar with advanced graph learning techniques.
Clarity for Non-experts: Some parts might need more explanation or simplification for readers who aren't experts in the field.
Generalizability: While the method looks promising, the paper doesn't discuss how it might apply to other domains or datasets beyond those tested.

In summary, the paper makes a significant contribution to the field of cognitive diagnosis in intelligent education, offering a new method that tackles key challenges like semantic preservation and data sparsity.

---

### Official Review · Reviewer_Noav · 2025-01-20
**Semantic-tailored Variational-Contrastive Graph Learning for Cognitive Diagnosis**

**Rating:** 9
**Confidence:** 5

**Review:**

**Summary**

This paper delves into cognitive diagnosis through graph contrastive learning (GCL) to address the limitations of data sparsity. The authors identify two critical challenges in existing GCL-based methods: disrupting the relationship in response graph and ignoring semantic heterogeneity between correct and incorrect nodes. For this, this paper proposes a semantic-tailored variational-contrastive graph cognitive diagnosis by introducing a semantic specific variational graph reconstruction module with a semantic-specific contrastive learning strategy. Extensive experiments demonstrate the effectiveness of the proposed method.

**Pros**

1.	Good Writing. This paper is well-written with a clear and concise presentation. I can easily follow the storyline.
2.	Novel Method. The authors identify the critical limitations of GCL-based methods and design a novel self-supervised cognitive method SVGCD.
3.	Adequate Experiments. This paper provides a thorough and comprehensive experimental comparison, demonstrating the reproducibility and validity of the method.

**Cons**

As stated in Introduction, existing methods often overlook the impact of data sparsity. How to categorize the students into sparse, medium and dense groups?

---

### Meta-Review · Area_Chair_LTRo · 2025-01-26

**Recommendation:** Accept (Oral)
**Confidence:** 1

**Metareview:**

This paper is very well-researched, outlines a real-world problem, and proposes a novel model to address CD. The clarity of the paper is very easy to follow and provides a clear presentation including architecture diagrams and modular explanations. This paper also offers real-world datasets rigorous baselines. Though only two datasets are used, this paper provides a novel combination of semantic-aware learning for CD is innovative by tacking data sparsity challenge.

Recommendation: Strong contribution to the field of CD with self-supervised learning.

---

### Decision · Program_Chairs · 2025-01-26

**Decision:**

Accept (Oral)

**Comment:**

The program chair concurs with the area chair's decision.

For the camera-ready version, please revise your paper according to the feedback provided by the reviewers.

Workshop papers must be written in English, follow a double-column format, and comply with the [ACM template](https://www2025.thewebconf.org/short-papers) and formatting guidelines. The template is also available in [Overleaf](https://www.overleaf.com/latex/templates/association-for-computing-machinery-acm-sig-proceedings-template/bmvfhcdnxfty). For authors using Microsoft Word, the Word Interim Template is recommended.

Camera-ready versions of accepted papers can and should include all information to identify authors, and should acknowledge any funding received that directly supported the presented research.

In addition, ensure that the DOI (to be provided by the PCs at a later stage) is included, and cite the workshop (to appear) using the following reference:

```
@inproceedings{time2025,
  title={TIME 2025: 1st International Workshop on Transformative Insights in Multi-faceted Evaluation},
  author={Lei Wang and Md Zakir Hossain and Syed Islam and Tom Gedeon and Sharifa Alghowinem and Isabella Yu and Serena Bono and Xuanying Zhu and Gennie Nguyen and Nur Haldar and Seyed Jalali and Abdur Razzaque and Imran Razzak and Rafiqul Islam and Shahadat Uddin and Naeem Janjua and Aneesh Krishna and Manzur Ashraf},
  booktitle={ACM Web Conference Workshop},
  year={2025}
}
```

Please note that at least one in-person registration is required for each accepted workshop paper to be included in the Companion Proceedings of WWW 2025. All accepted papers must be presented at the conference. Papers not presented (no-shows) may be withdrawn from the companion proceedings. Presentations will be conducted in two formats: oral and poster.

The camera-ready deadline for workshop papers is 7 February 2025 (AoE).